# Influence of Pneumonia on the Survival of Patients with COPD

**DOI:** 10.3390/jcm9010230

**Published:** 2020-01-15

**Authors:** Zichen Ji, Julio Hernández Vázquez, José María Bellón Cano, Virginia Gallo González, Beatriz Recio Moreno, Alicia Cerezo Lajas, Luis Puente Maestu, Javier de Miguel Díez

**Affiliations:** 1Pulmonology Service, Gregorio Marañón University General Hospital, 28007 Madrid, Spain; vgallogonzalez@gmail.com (V.G.G.); beatriz_remo@outlook.es (B.R.M.); naremaal_4@hotmail.com (A.C.L.); luis.puente@salud.madrid.org (L.P.M.); jdemigueldiez@telefonica.net (J.d.M.D.); 2Pulmonology Section, Infanta Leonor University Hospital, 28031 Madrid, Spain; juliohernandezvazquez@hotmail.com; 3Research Support Service, Gregorio Marañón Health Research Institute, 28007 Madrid, Spain; bellon23@gmail.com

**Keywords:** chronic obstructive pulmonary disease (COPD), respiratory infections, pneumonia, survival

## Abstract

Background: Pneumonia is a frequent infection. Chronic obstructive pulmonary disease (COPD) can present with comorbidities, including pneumonia. It is known that COPD worsens the evolution of pneumonia, but few studies describe the impact of pneumonia on COPD evolution. This study analyzes the influence of pneumonia on the survival of COPD patients. Methods: Observational study of a cohort of 273 patients with COPD who attended spirometry in 2011, with a prospective follow-up of six years. Patients were divided into two groups according to their acquisition of pneumonia during follow-up. The difference in survival between the two groups was analyzed. Results: Survival was lower in the group with pneumonia compared with that without pneumonia (*p* = 0.000), both globally and after stratification by COPD phenotype. Pneumonia (Hazard Ratio -HR- 2.65; 95% Confidence Interval -CI- 1.57–4.48), advanced age (HR 1.08; 95% CI 1.03–1.09), and high Charlson index (HR 1.31; 95% CI 1.17–1.47) were identified as risk factors independently associated with mortality, while a high body mass index (HR 0.92; 95% CI 0.87–0.96) was identified as a protective factor. Conclusions: Pneumonia is associated with worse prognosis in COPD patients. It is important to take into account this comorbidity for a comprehensive care of these patients.

## 1. Introduction

Pneumonia is a very common infection [1], with a worldwide annual incidence of 1.6–13.4 cases per 100,000 persons. It is also one of the most common reasons for admission to hospital, being an infectious disease [2,3]. Admissions for pneumonia generate considerable health care expenditure. The mean hospital stay of patients affected by pneumonia is around 7 days [4], and the direct cost of each admission is more than €1500 in Spain [5].

Chronic obstructive pulmonary disease (COPD) is a highly prevalent disease with elevated morbidity and mortality [6,7,8]. It is often accompanied by various comorbid conditions including arthritis, heart disease, arterial hypertension, diabetes mellitus, dyslipidemia, psychiatric disease, gastrointestinal disorders, cancer, and osteoporosis [9,10].

The clinical course of COPD is characterized by exacerbations, which are defined as a sharp change in baseline status (grade of dyspnea, cough, and expectoration) that surpasses the daily variability and requires treatment to be modified [11]. Pneumonia was previously thought to be a cause of COPD exacerbation [12], although it was recently shown that the characteristics of patients with COPD and pneumonia differ from those of patients without pneumonia who experience COPD exacerbation [13]. Consequently, pneumonia has come to be considered an additional comorbidity of COPD [11].

Various studies have shown that the presence of COPD worsens the course of patients with pneumonia [14,15], although few studies have focused on patients with COPD and thus have shown the effect of pneumonia on this disease. In this sense, Pasquale et al. [16] studied symptoms and quality of life in patients diagnosed with COPD who had pneumonia, although they did not analyze data on the consumption of health care resources and mortality resulting from pneumonia.

The objective of our study was to describe the characteristics of patients diagnosed with COPD who developed or did not develop pneumonia during follow-up. We also analyzed the effect of pneumonia on the clinical course and mortality of these patients.

## 2. Patients and Methods

### 2.1. Design

We performed an observational study in a hospital in Madrid, Spain. We collected data on patients with a confirmed previous diagnosis of COPD. Patients were older than 40 years, had a smoking history of ≥10 packet-years, and had attended the clinic to undergo spirometry between January and June 2011. Their forced expiratory volume in the first second (FEV1) was <70% of the predicted value. We excluded patients with a previous diagnosis of nonobstructive chronic respiratory diseases and those who were participating in clinical trials.

In order to avoid selection bias, patients were included in the order they attended the clinic. To be included, they had to fulfill all of the inclusion criteria and none of the exclusion criteria.

### 2.2. Variables Collected

Variables were collected from medical records in 2 phases. Patient data were collected at inclusion, and data on development of pneumonia and mortality were collected during follow-up.

The variables collected during the inclusion phase were age, sex, weight, height, body mass index (BMI), date of inclusion in the study, results of spirometry, active smoking, phenotype, treatment of COPD, respiratory therapy, and comorbid conditions included in the Charlson comorbidity index and the COPD-specific comorbidity test (COTE).

The variables collected during follow-up were date of the first episode of pneumonia, number of episodes of pneumonia, last follow-up visit, and date of death.

An episode of pneumonia was considered as such if it was managed in the hospital with a discharge report showing a diagnosis of community-acquired or nosocomial pneumonia and radiology data that were consistent with this diagnosis.

We took into account 4 COPD phenotypes: positive bronchodilator response, nonexacerbator, exacerbator with emphysema, and exacerbator with chronic bronchitis. Patients were classified into these 4 phenotypes by first including all those with an increase in FEV1 greater than 12% and 200 mL during the bronchodilation test in the positive bronchodilator response phenotype. Next, patients who had not made at least 2 visits to the emergency department with an exacerbation of COPD or had at least 1 episode of exacerbation with admission to hospital during the year before inclusion were included in the nonexacerbator phenotype. Among the remaining patients, we classed as exacerbators with emphysema those whose predominant symptom was dyspnea or who had radiological data indicating emphysema or a low diffusing capacity for carbon monoxide (DLCO). Finally, patients who fulfilled the criteria for chronic bronchitis, that is, cough and expectoration for at least 3 months per year during 2 consecutive years, were classified as exacerbators with chronic bronchitis.

Patients were followed up until 1 April 2017. If a patient died during follow-up, the end of follow-up was the date of death; if the patient survived, the end of follow-up was the last date on which the patient was seen by the health service.

### 2.3. Statistical Analysis

We verified whether the distribution of the variables was normal using a histogram or, depending on whether the sample size was less than or greater than 50, using the Kolmogorov–Smirnov test or the Shapiro–Wilk test, respectively.

Normally distributed quantitative variables were expressed as mean (*SD*). Non-normally distributed quantitative variables were expressed as median (IQR). Qualitative variables were expressed as frequencies.

Quantitative variables were compared using an analysis of variance or the Kruskal–Wallis test, depending on whether the distribution was normal. Qualitative variables and proportions were compared using the Chi-squared test or Fisher exact test, depending on the sample size.

Survival was analyzed using a Kaplan–Meier plot, and the log-rank test was used for the univariate comparison of the probability of death depending on the presence of pneumonia during follow-up. Cox regression was used for multivariate analysis after adjusting for age, sex, BMI, lung function, active smoking, phenotype, and Charlson comorbidity index.

Acquisition of pneumonia was considered a time-dependent variable. Proportionality of risk was also verified.

Statistical significance was set at *p* < 0.05 (2-tailed) for all comparisons.

The statistical analysis was performed using SPSS for Windows, Version 20, and Stata Version 14.

### 2.4. Ethics Committee and Informed Consent

The study was approved by the Ethics Committee of the Hospital General Universitario Gregorio Marañón. Patients gave their signed informed consent before participating in the study.

## 3. Results

The study population comprised 273 patients, of whom 243 were men (89%). Mean age was 67.99 (10.62) years, mean weight was 75.03 kg (16.89), mean height was 1.63 m (0.08), and mean BMI was 28.05 (5.49) kg/m^2^. The median follow-up was 68.16 (40.69–72.12) months. As for lung function at inclusion, the mean FEV1 was 48.64% of the predicted value (12.59), and the mean forced vital capacity (FVC) was 73.18% of the predicted value (15.00). At baseline, 92 patients (34%) were active smokers. By phenotype, 71 patients (26.0%) were classified in the positive bronchodilator response phenotype, 135 (49.5%) as nonexacerbators, 27 as exacerbators with emphysema (9.9%), and the remaining 40 patients as exacerbators with chronic bronchitis (14.7%). As for comorbidities, the median Charlson comorbidity index was 2 (1–4), and the median COTE score was 1 (0–2). As for treatment, 254 (93%) patients were treated with a long-acting muscarinic antagonist, 242 patients (88.6%) with a long-acting β_2_-adrenergic agonist, and 212 patients (77.7%) with an inhaled corticosteroid. With regard to respiratory therapy, 91 patients (33.3%) received long-term home oxygen therapy, 31 (11.4%) were treated with continuous positive airway pressure, and 14 (5.1%) with bilevel positive airway pressure. Before inclusion in the study, 164 (60.1%) patients were vaccinated against pneumococcus. During follow-up, 77 patients (28.2%) had at least one episode of pneumonia, and 93 patients (34.1%) died. The complete descriptive data are shown in Table 1.

Table 2 shows comparisons of anthropometric data, lung function, and phenotype between groups with and without pneumonia during follow-up. Statistically significant differences were found for inhaled corticosteroids, home oxygen therapy, and pneumococcal vaccination. Patients were older and had poorer lung function in the pneumonia group. In addition, patients with pneumonia received inhaled corticosteroids and long-term home oxygen therapy more often than those without pneumonia.

Comorbidities were compared both at the individual and at the group level according to the Charlson comorbidity index and COTE scale between patients with and without pneumonia during follow-up. Statistically significant differences were found for dyslipidemia and heart failure (Table 3).

The overall survival curve according to whether patients had or not pneumonia during follow-up is shown in Figure 1. Survival was also stratified by phenotype (Figure 2). Statistically significant differences were found for all comparisons. Survival was poorer in the pneumonia group, except for the exacerbator-with-chronic-bronchitis phenotype, for which no statistically significant differences were found. The factors that were independently associated with mortality are shown in Table 4. The independent risk factors for mortality were acquisition of pneumonia, older age, and high Charlson comorbidity index. BMI, on the other hand, was an independent protective factor for mortality.

## 4. Discussion

The main finding of our study is that pneumonia in patients with COPD is associated with poorer survival, both overall and in patients with the phenotypes positive bronchodilator response, nonexacerbator, and exacerbator with emphysema. This association was not found in patients with the exacerbator-with-chronic-bronchitis phenotype. Patients with pneumonia were more often older, more frequently had dyslipidemia and heart failure, and more frequently received treatment with inhaled corticosteroids and long-term home oxygen therapy than patients who did not have pneumonia. The presence of more comorbidities was not associated with a greater probability of pneumonia, although it was an independent risk factor for mortality.

COPD progresses with comorbid conditions [11,17], which are produced through various mechanisms [9]: common risk factors, local and systemic inflammation leading to COPD, and possibly as yet unknown common genetic factors. The main national and international guidelines recommend a systematic evaluation of comorbid conditions in patients with COPD, since these can affect disease progression both in the stable phase and during exacerbations [11,18].

As mentioned above, pneumonia is currently considered an additional comorbidity of COPD [13]. In this case, the association between the two diseases arises from the fact that they share risk factors, such as smoking, environmental pollution, older age, and immunosuppression [19].

As for the effect of pneumonia in COPD and vice versa on mortality, study findings are contradictory. In their study on community-acquired pneumonia (CAP) in patients with and without COPD, Bonnesen et al. [14] concluded that the presence of COPD is not a risk factor for mortality at 30 days in patients who have had an episode of community-acquired pneumonia. In contrast, Sharafkhaneh et al. [15], who performed a similar study, proposed that pneumonia and COPD affect short- and long-term mortality differently, with pneumonia having a greater short-term effect, and COPD a greater long-term effect.

Our results show that pneumonia does in fact affect mortality in patients with COPD. However, it is noteworthy that the decrease in survival begins to occur at 18 months from the first episode of pneumonia. This finding is consistent with those of the two previously cited studies [14,15], since there was no association between pneumonia and increased mortality at 30 days, although an association was found in the long term, possibly owing to COPD itself more than to pneumonia. This observation could be explained by the possibility that, despite being an acute disease, pneumonia leads to the deterioration of symptoms and quality of life in patients with COPD [16], which may not be fully recovered after resolution of pneumonia, thus leading to poorer long-term progress.

The difference in mortality was demonstrated for different phenotypes, although it is striking that no statistically significant differences were found for the exacerbator-with-chronic-bronchitis phenotype, possibly because this phenotype is associated with poorer survival than the other phenotypes [8]. In fact, we found that the Kaplan–Meier survival curve of patients with pneumonia having this phenotype was much lower than that of the group of patients without pneumonia at 18 months presenting the other phenotypes. However, the distance between the curves decreased in the long term.

Inhaled corticosteroids form part of the therapeutic regimen of COPD and are indicated in patients with COPD who have a previous diagnosis of asthma, in those with peripheral blood eosinophilia or a very positive bronchodilation response, and in those who require a triple therapy for the control of their disease [11,18]. Despite the usefulness of inhaled corticosteroids, their adverse effects are well known, the most frequent being pneumonia, oropharyngeal candidiasis, and osteoporotic fractures [20]. Previous studies have shown that patients with pneumonia more frequently take inhaled corticosteroids [21,22], and our results are consistent with this observation. Nevertheless, other authors also reported contradictory results [23]. Furthermore, inhaled corticosteroids, which are indicated for patients with COPD, are often overprescribed [22,24,25,26,27]—as observed in the present study—thus potentially increasing the risk of pneumonia.

Once again, the present study confirms that advanced age and presence of comorbid conditions (i.e., higher Charlson comorbidity index) are independent risk factors associated with mortality in patients with COPD [9,10,28,29].

Lastly, mention must be made of the “obesity paradox” in COPD, which involves a high BMI acting as a protective factor for mortality in affected patients, whereas in the vast majority of diseases, it is usually a risk factor. While the origin of this paradox is unknown, several hypotheses have been proposed, including the notion that BMI is not a good descriptor of fatty mass and lean mass or that spirometry overestimates the degree of obstruction in obese patients [30,31,32].

Our study has both strengths and limitations. We included patients who came from different departments to undergo spirometry. Therefore, the sample is representative of the population of patients with COPD in real life. Nevertheless, this means that we were only able to record data on lung function, comorbidities, and classification of the phenotype at inclusion. The prospective follow-up was performed by means of a review of the patients’ clinical history; therefore, the analysis was performed using baseline data, with no adjustment for episodes of pneumonia and death according to the patients’ clinical situation at the time of the episode. The study design enabled us to determine the effect of pneumonia on mortality independently of the course of COPD, although in order to ascertain this effect at different points during the course of the disease, it would be necessary to use a design that enables us to follow patients in person in order to obtain data on their clinical progress. Due to this limitation derived from the study design, the results and conclusions can only refer to the influence of pneumonia on patients with COPD in general and cannot evaluate the differences of this influence according to the severity of the disease. In addition, this study was designed following the classification of patients with COPD proposed by the Spanish guidelines [11], which allows classification by phenotypes. We do not have data on the GOLD classification, which takes into account symptoms and exacerbations, so it was not been possible to analyze the mortality of these patients from this point of view. Another limitation presented by the study is the inequality of the sample size of each phenotype, because the cohort consisted of outpatients. In this way, we had more patients with a nonexacerbator phenotype and fewer patients with the other phenotypes. This may have decreased the statistical power of the analysis for certain phenotypes, for example, the exacerbator-with-chronic-bronchitis phenotype.

## 5. Conclusions

We found that having pneumonia was associated with poorer long-term survival in patients with COPD, both overall and by phenotype, except for the exacerbator-with-chronic-bronchitis phenotype. Advanced age and presence of comorbidities were identified as independent risk factors for mortality, whereas BMI acted as a protective factor. Pneumonia should be considered an additional comorbid condition in the integrated care of patients with COPD. COPD patients’ management should include a careful assessment of the individual risk/benefit ratio of inhaled corticosteroids, since the onset of pneumonia is indicative of a poorer prognosis in patients with COPD.

## Figures and Tables

**Figure 1 jcm-09-00230-f001:**
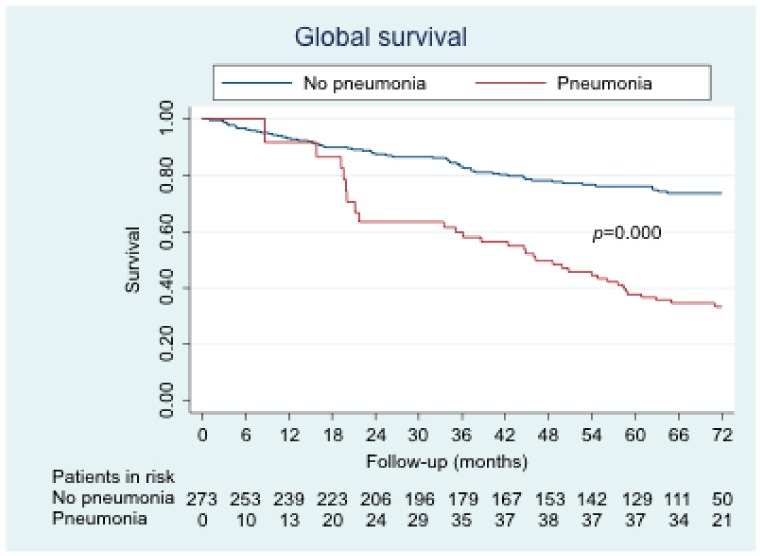
Overall survival according to the presence or absence of pneumonia.

**Figure 2 jcm-09-00230-f002:**
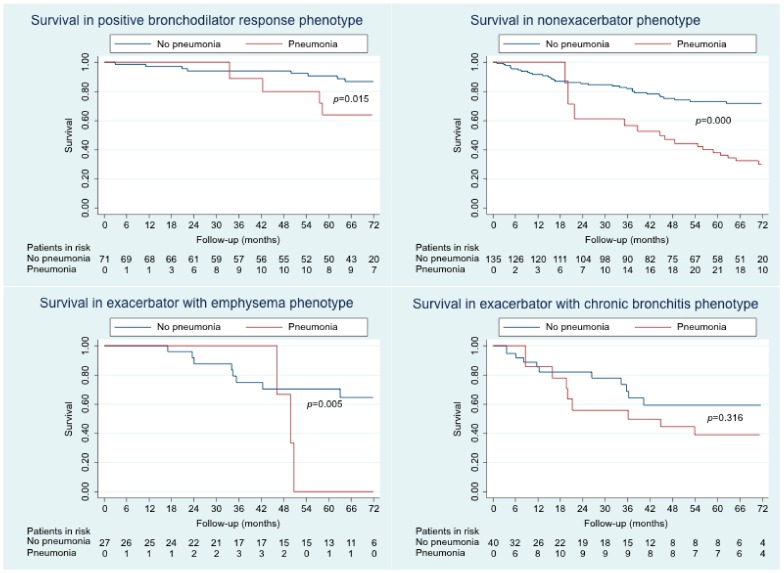
Survival according to the presence or absence of pneumonia in each COPD phenotype.

**Table 1 jcm-09-00230-t001:** General characteristics of the studied patients.

Variable	All Phenotypes	Positive Bronchodilator Response	Non-Exacerbator	Exacerbator with Emphysema	Exacerbator with Chronic Bronchitis
Patients, *n* (%)	273 (100.0)	71 (26.0)	135 (49.5)	27 (9.9)	40 (14.7)
Follow-up, months (IQR)	68.15 (40.69–72.12)	70.99 (54.13–73.45)	66.55 (37.83–71.74)	61.32 (35.33–71.94)	41.65 (20.18–71.21)
Males, *n* (%)	243 (89.0)	63 (88.7)	124 (91.8)	22 (81.5)	34 (85.0)
Age, years (*SD*)	67.99 (10.62)	63.44 (11.70)	69.76 (9.33)	67.44 (11.23)	70.47 (10.10)
Weight, kg (*SD*)	75.03 (16.89)	77.01 (16.94)	76.04 (17.75)	71.85 (18.46)	66,78 (10.85)
Height, m (*SD*)	1.63 (0.08)	1.65 (0.09)	1.63 (0.07)	1.61 (0.08)	1.61 (0.08)
BMI, kg/m^2^ (*SD*)	28.05 (5.49)	28.15 (5.44)	28.48 (5.83)	27.46 (6.19)	26.82 (3.55)
FEV_1_, % (*SD*)	48.64 (12.59)	53.71 (11.66)	48.57 (12.52)	44.66 (11.82)	42.53 (11.55)
FVC, % (*SD*)	73.18 (15.00)	79.60 (13.19)	71.74 (15.08)	71.58 (14.09)	67.72 (15.11)
Active smoking, *n* (%)	92 (34%)	35 (49.3)	36 (26.7)	7 (25.9)	14 (35.0)
Comorbidity indexes, median (IQR)					
Charlson	2 (1–4)	2 (1–3)	2 (1–4)	2 (1–3)	3 (1–4)
COTE	1 (0–2)	0 (0–2)	1 (0–3)	0 (0–3)	1 (0–3)
Pharmacological treatment, *n* (%)					
LAMA	254 (93.0)	60 (84.5)	129 (95.6)	25 (92.6)	40 (100.0)
LABA	242 (88.6)	66 (93.0)	111 (82.2)	26 (96.3)	39 (97.5)
ICS	212 (77.7)	61 (84.5)	91 (67.4)	23 (85.2)	37 (92.5)
Respiratory therapies, *n* (%)					
LTOT	91 (33.3)	14 (20.0)	39 (28.9)	13 (48.1)	25 (62.5)
CPAP	31 (11.4)	10 (14.1)	16 (11.9)	3 (11.1)	2 (5.0)
BiPAP	14 (5.1)	1 (1.4)	6 (4.4)	2 (7.4)	5 (12.5)
Pneumococcal vaccination, *n* (%)	160 (58.6)	32 (45.1)	86 (63.7)	16 (59.3)	26 (65.0)
Pneumonia, *n* (%)	77 (28.2)	15 (21.1)	38 (28.1)	5 (18.5)	19 (47.5)
Death, *n* (%)	93 (34.1)	12 (16.9)	49 (36.3)	12 (44.4)	20 (50.0)

IQR: interquartile range; *SD*: standard deviation; BMI: body mass index; FEV_1_: forced expiratory volume in the first second; FVC: forced vital capacity; COTE: chronic obstructive pulmonary disease (COPD)-specific comorbidity test; LAMA: long-acting muscarinic antagonist; LABA: long-acting β_2_-adrenergic agonist; ICS: inhaled corticosteroid; LTOT: long-term home oxygen therapy; CPAP: continuous positive airway pressure; BiPAP: bilevel positive airway pressure.

**Table 2 jcm-09-00230-t002:** Anthropometric, lung function, phenotype, and treatment data according to the presence or absence of pneumonia.

Variable	Pneumonia	No Pneumonia	*p*-Value
Male, *n* (%)	69 (89.6)	174 (88.8)	0.843
Age, years (*SD*)	71.65 (8.90)	66.55 (10.91)	0.000 ***
Weight, kg (*SD*)	75.06 (15.20)	75.02 (17.52)	0.984
Height, m (*SD*)	1.62 (0.08)	1.63 (0.80)	0.140
BMI, kg/m^2^ (DE)	28.53 (5.35)	27.87 (5.55)	0.369
Active smoking, *n* (%)	21 (27.3)	71 (36.2)	0.200
Pulmonary function, % (*SD*)			
FEV_1_	47.37 (13.20)	49.14 (12.34)	0.297
FVC	70.81 (14.82)	74.11 (15.01)	0.102
Phenotypes, *n* (%)			
Positive bronchodilator response	15 (19.5)	56 (28.6)	0.016 *
Nonexacerbator	38 (49.4)	97 (49.5)	
Exacerbator with emphysema	5 (6.5)	22 (11.2)	
Exacerbator with chronic bronchitis	19 (24.7)	21 (10.7)	
Pharmacological treatment, *n* (%)			
LAMA	75 (97.4)	179 (91.3)	0.076
LABA	71 (92.2)	171 (87.2)	0.245
ICS	66 (85.7)	146 (74.5)	0.045 *
Respiratory therapies, *n* (%)			
LTOT	34 (44.2)	57 (29.1)	0.017 *
CPAP	5 (6.5)	26 (13.3)	0.139
BiPAP	6 (7.8)	8 (4.1)	0.229
Pneumococcal vaccination, *n* (%)	21 (27.3)	139 (70.9)	0.000 ***

* *p* < 0.05; *** *p* < 0.001.

**Table 3 jcm-09-00230-t003:** Comorbidities according to the presence or absence of pneumonia.

Comorbidities, *n* (%)	Pneumonia	No Pneumonia	*p*-Value
Arterial hypertension	41 (53.2)	105 (53.6)	0.961
Dyslipidemia	40 (51.9)	68 (34.7)	0.009 **
Mellitus diabetes	22 (28.6)	44 (22.4)	0.288
Atrial fibrillation	13 (16.9)	22 (11.2)	0.229
Ischemic heart disease	14 (18.2)	24 (12.2)	0.243
Heart failure	17 (22.1)	20 (10.2)	0.017 *
Peripheral vasculopathy	5 (6.5)	6 (3.1)	0.302
Cerebrovascular disease	0 (0.0)	1 (0.5)	1.000
Dementia	1 (1.3)	4 (2.0)	1.000
Connective tissue disease	2 (2.6)	1 (0.5)	0.193
Gastroduodenal ulcer	6 (7.8)	8 (4.1)	0.229
Liver disease	4 (5.2)	12 (6.1)	1.000
Hemiplegia	0 (0.0)	0 (0.0)	-
Chronic kidney disease	9 (11.7)	14 (7.1)	0.232
All neoplasia	19 (24.7)	56 (28.6)	0.550
Lung cancer	6 (7.8)	18 (9.2)	0.816
AIDS	1 (1.3)	0 (0.0)	0.282
Anxiety	0 (0.0)	4 (2.0)	0.580
Idiopathic pulmonary fibrosis	0 (0.0)	0 (0.0)	-
Cirrhosis	0 (0.0)	0 (0.0)	-
Comorbidity indices, median (IQR)			
Charlson	2 (1–4)	2 (1–3)	0.099
COTE	1 (1–3)	0 (0–2)	0.244

AIDS: acquired immune deficiency syndrome; * *p* < 0.05; ** *p* < 0.01.

**Table 4 jcm-09-00230-t004:** Risk factors independently associated with mortality.

Variable	Hazard Ratio	95% CI	*p*-Value
Pneumonia	2.65	1.57–4.48	0.000 ***
Age	1.08	1.03–1.09	0.000 ***
Male	1.95	0.74–5.15	0.180
BMI	0.92	0.87–0.96	0.001 **
FEV_1_	1.00	0.97–1.02	0.780
FVC	0.99	0.97–1.00	0.142
Active smoking	1.29	0.76–2.19	0.353
Phenotypes			
Exacerbator with emphysema	2.27	0.97–5.33	0.060
Exacerbator with chronic bronchitis	2.12	0.97–4.62	0.059
Nonexacerbator	1.68	0.86–3.27	0.129
Pneumococcal vaccination	0.92	0.58–1.45	0.720
Charlson	1.31	1.17–1.47	0.000 ***

CI: confidence interval; ** *p* < 0.01; *** *p* < 0.001.

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
