# Peer review of "Influence of Pneumonia on the Survival of Patients with COPD"

_jcm, 2020, doi:10.3390/jcm9010230_

Round 1
Reviewer 1 Report
Authors wished to study the influence of pneumonia on the
survival of COPD patients. At this aim they performed an observational study of a cohort of 273
patients with COPD with a prospective follow-up of 6 years. They found that survival was lower in the group with pneumonia compared to whose without pneumonia . Because Authors did not observe any association between pneumonia and increased mortality at 30 days, they suggest that COPD itself may explain the higher mortality. Advanced age and high Charlson index
were identified as risk factors independently associated with mortality, while a high body mass index was identified as a protective factor.
Minor points
-I suggest to investigate whether classifying patients according to Gold (I-IV) and dyspnea/exacerbation rate (A-D) could explain the higher mortality of patients with pneumonia
-Information about pneumococcal vaccines should be reported
Author Response
Point 1: I suggest to investigate whether classifying patients according to Gold (I-IV) and dyspnea/exacerbation rate (A-D) could explain the higher mortality of patients with pneumonia.

Response 1: This study was designed following the classification of patients with COPD proposed by the Spanish guidelines, which allows classification by phenotypes. Unfortunately, we did not collect information about the GOLD classification. We add this limitation in the discussion section of the new version of the manuscript adding the following comment:
“In addition, this study was designed following the classification of patients with COPD proposed by the Spanish guideline, which allows classification by phenotypes. We do not have data on the GOLD classification, which takes into account the symptoms and exacerbations, so it has not been possible to analyze the mortality of these patients from this point of view.”
Point 2: Information about pneumococcal vaccines should be reported.
Response 2: Taking into account the reviewer’s comment, we have added the following comment in the results section and in the table 1:
“Before inclusion in the study, 164 (60.1%) patients were vaccinated against pneumococcus.”

Reviewer 2 Report
Jcm-682624-sub - Reviewer’s comments
Overall comments: This manuscript entitled “Influence of pneumonia on the survival of patients with COPD” investigates the relationship between COPD, pneumonia and mortality, which is clinically relevant. This study shows a significant relationship between presence of pneumonia and worsened prognosis in COPD. I would recommend this manuscript a major revision. This manuscript could also use a thorough proofread for spelling and grammatical errors; for example, there are errors in table 1&2 and the conclusion paragraphs. Also, review the paragraph structure. I am always concerned when there are several paragraphs containing one sentence. Consider using subheadings.
Specific comments
Abstract
The conclusion statement in this sections is not substantiated with the evidence. You have not investigated causality here, simply the relationship between the two variables. I would suggest altering this statement.
Introduction
Nice rationale for the study.
Methods
Where was this data collected from? Medical records or was it self-reported from the patient?
Results
There was a very high proportion of male patients (89%). I can see that there was no significant difference in the distribution of sex between the pneumonia and non-pneumonia groups, and the hazard ratio was not significantly increased, indicating that is wasn’t an independent risk factor, which is contradictory to previous research (eg McGhan 2007 Chest). I would like to see a more in-depth of the impact of gender on the risk of mortality. What are the proportions within the phenotypes? Is gender an independent risk factor within certain phenotypes? Etc. There is no mention of COPD severity in this study. COPD severity increased the risk of hospitalisations and re-administration, which increases the risk of pneumonia and mortality. I highly recommend investigating the relationship between the study outcomes and COPD severity. In the co-morbidities table (table 3) there’s no mention of cancer. COPD is a major risk factor for lung cancer. I highly recommend including previous cancer diagnosis on the comorbidities table and specifically lung cancer as a separate co-morbidity, as these are major co-morbidities, which would affect mortality. There is no data on the cause of death for these patients. If you conclude that pneumonia causes a worse prognosis for COPD, this would be demonstrated in the cause of death. The active smokers - did any quit smoking throughout the follow-up period? This would have affected the risk of co-morbidities and death. I’d like to see a sub-group cohort description of the general and clinical demographics for the different phenotypic cohorts, in addition to table 1, 2 and 3, rather than only having the collated cohort. It would be good so see how these different factors are represented across the cohorts. Figure 2 shows no significant difference in survival for the exacerbator with chronic bronchitis. There was a significant difference for all other phenotype. However is you look at the data presented in Table 2, the sample size differ greatly in these phenotypes, whereas these is a similar comparative samples sizes for the exacerbator with chronic bronchitis phenotype. This finding should be validate in a larger cohort, or a cohort with similar samples sizes in each subgroup. Otherwise this limitation should be discussed in depth in the discussion section.
Discussion
The authors have stated that the study has both strengths and limitations, yet fails to discuss the limitations in this discussion and the impact of the results/conclusions. I highly suggest adding a limitations section to the discussion. In the conclusion paragraph, it was stated that COPD was a frequent cause of death for the chronic bronchitis phenotype, yet the cause of death data was not presented in the manuscript. I would suggest including this data and analysis.
Author Response
Point 1: This manuscript could also use a thorough proofread for spelling and grammatical errors; for example, there are errors in table 1&2 and the conclusion paragraphs. Also, review the paragraph structure. I am always concerned when there are several paragraphs containing one sentence. Consider using subheadings.
Response 1: In the new version of the manuscript we have reviewed and corrected the errors and the paragraph structure.
Point 2: Abstract: The conclusion statement in this section is not substantiated with the evidence. You have not investigated causality here, simply the relationship between the two variables. I would suggest altering this statement.
Response 2: Taking into account the reviewer’s comment, we correct the wording of the conclusion in the new version of the manuscript:
“Conclusions: Having pneumonia is associated with worse prognosis in COPD patients. It’s important to take into account this comorbidity in comprehensive care for these patients.”
Point 3: Methods: Where was this data collected from? Medical records or was it self-reported from the patient?
Response 3: The date was collected from medical records. We add this information in the new version of the manuscript:
“Variables were collected from medical records in 2 phases.”
Point 4: There was a very high proportion of male patients (89%). I can see that there was no significant difference in the distribution of sex between the pneumonia and non-pneumonia groups, and the hazard ratio was not significantly increased, indicating that is wasn’t an independent risk factor, which is contradictory to previous research (eg McGhan 2007 Chest). I would like to see a more in-depth of the impact of gender on the risk of mortality. What are the proportions within the phenotypes? Is gender an independent risk factor within certain phenotypes?
Response 4: Taking into account the reviewer’s comment, we have included gender data according to each phenotype in the table 1. Gender and phenotypes were already included in the multivariate model, and gender was not an independent risk factor considering the phenotypes.
Point 5: There is no mention of COPD severity in this study. COPD severity increased the risk of hospitalisations and re-administration, which increases the risk of pneumonia and mortality. I highly recommend investigating the relationship between the study outcomes and COPD severity.
Response 5: The severity of COPD, in part, depends on lung function. In this sense, FEV1 is included in the analysis in this study, both in the descriptive part (table 1) and in the multivariate model (table 4).
Point 6: In the co-morbidities table (table 3) there’s no mention of cancer. COPD is a major risk factor for lung cancer. I highly recommend including previous cancer diagnosis on the comorbidities table and specifically lung cancer as a separate co-morbidity, as these are major co-morbidities, which would affect mortality.
Response 6: Taking into account the reviewer’s comment, we have extended table 3 in the new version of the manuscript with the data of lung cancer separately.
Point 7: There is no data on the cause of death for these patients. If you conclude that pneumonia causes a worse prognosis for COPD, this would be demonstrated in the cause of death.
Response 7: We collected information on patient survival through a joint clinical history of the Madrid health service. In the case that the patient dies in a center other than ours, we cannot know the cause of death, but only the date of the same. We believe that, in a large cohort, the different causes of death should be homogeneously distributed between the two comparison groups. In addition, as mentioned in the fifth paragraph of the discussion section, the reduction in survival is observed mainly after 18 months of having pneumonia, that is, we think that pneumonia is not necessarily the direct cause of death, but that it deteriorates the patient and worsens COPD.
Point 8: I’d like to see a sub-group cohort description of the general and clinical demographics for the different phenotypic cohorts, in addition to table 1, 2 and 3, rather than only having the collated cohort. It would be good so see how these different factors are represented across the cohorts.
Response 8: Taking into account the reviewer’s comment, we have extended table 1 in the new version of the manuscript with the data of each phenotype separately.
Point 9: Figure 2 shows no significant difference in survival for the exacerbator with chronic bronchitis. There was a significant difference for all other phenotype. However is you look at the data presented in Table 2, the sample size differ greatly in these phenotypes, whereas these is a similar comparative samples sizes for the exacerbator with chronic bronchitis phenotype. This finding should be validate in a larger cohort, or a cohort with similar samples sizes in each subgroup. Otherwise this limitation should be discussed in depth in the discussion section.
Response 9: Taking into account the reviewer’s comment, we have added the following comment in the discussion section:
“Another limitation presented by the study is the inequality of the sample size of each phenotype, because this cohort comes from outpatients. In this way, we have more patients with a nonexacerbator phenotype and fewer patients with the other phenotypes. This may cause the analysis to have less statistical power in certain phenotypes, for example, exacerbator with chronic bronchitis.”
Point 10: Discussion: The authors have stated that the study has both strengths and limitations, yet fails to discuss the limitations in this discussion and the impact of the results/conclusions. I highly suggest adding a limitations section to the discussion.
Response 10: Taking into account the reviewer’s comment, we have added the following comment in the discussion section:
“Due to this limitation derived from the design, the results and conclusions can only refer to the influence of pneumonia on patients with COPD in general, but we cannot know the differences of this influence according to the severity of the disease.”
Point 11: In the conclusion paragraph, it was stated that COPD was a frequent cause of death for the chronic bronchitis phenotype, yet the cause of death data was not presented in the manuscript. I would suggest including this data and analysis.
Response 11: We collected information on patient survival through a joint clinical history of the Madrid health service. In the case that the patient dies in a center other than ours, we cannot know the cause of death, but only the date of the same. To avoid confusion, we have removed the phrase that refers to the cause of death of exacerbator with chronic bronchitis phenotype patients with from the conclusions section.

Round 2
Reviewer 1 Report
Authors changed the revised manuscript according to the suggestions.
Authors state that 60% of their patients received pneumococcal vaccine, but they do not report the effect of vaccine on the risk of acquiring pneumonia as well on mortality. I would like this information to be reported in the final manuscript .
Author Response
Point 1: Authors changed the revised manuscript according to the suggestions.
Authors state that 60% of their patients received pneumococcal vaccine, but they do not report the effect of vaccine on the risk of acquiring pneumonia as well on mortality. I would like this information to be reported in the final manuscript.
Response 1: Taking into account the reviewer's comment, we incorporate data on pneumococcal vaccination in Table 2 to show its effect on the acquisition of pneumonia and we mention this statistically significant difference in the text of the results section.
“Statistically significant differences were found for inhaled corticosteroids, home oxygen therapy and pneumococcal vaccination.”
We also incorporate pneumococcal vaccination in the multivariate model, shown in Table 4, to reflect its relationship with mortality. By including a new variable, the other variables have undergone minor changes in their result, which we update in the table. No results have been different than the previous analysis.

Reviewer 2 Report
Thank you for your consideration of my comments/feedback. I am satisfied by the changes made by the authors.
Author Response
Point 1: Thank you for your consideration of my comments/feedback. I am satisfied by the changes made by the authors.
Response 1: Thank you for your review and your comments.
